# Numerical Study on Effect of Aggregate Moisture on Mixing Process

**DOI:** 10.3390/ma17040898

**Published:** 2024-02-15

**Authors:** Guodong Cao, Sheng Xie, Daiqiang Deng, Shengqiang Jiang

**Affiliations:** College of Civil Engineering, Xiangtan University, Xiangtan 411105, China; 202121572160@smail.xtu.edu.cn (S.X.); xjddq@xtu.edu.cn (D.D.); jsqcx@xtu.edu.cn (S.J.)

**Keywords:** concrete, mixing, moisture content, angle of repose, liquid bridge force, discrete element method

## Abstract

During the concrete mixing process, the transition of aggregates from a dry to a moist state introduces a crucial dynamic that significantly influences particle interaction, consequently impacting mixing homogeneity. In this paper, based on the discrete element method, the effect of aggregate moisture on the mixing process of sand and stone was investigated. The interaction between dry particles was described by the Hertz–Mindlin model, while the interaction between wet particles was calculated by the linear cohesion model considering the liquid bridge force. Additionally, a functional relationship between the moisture content and the parameters of the linear cohesive contact model was established. The results show that the numerical method can be employed to simulate the mixing process. Notably, when the moisture content of pebbles ranges from 0% to 0.75% and that of sand ranges from 0% to 10.9%, the linear cohesion model is deemed suitable. The standard deviation of the mixing homogeneity of wet particles is lower than that of dry particles for short mixing time, indicating that a small amount of liquid enhances mixing homogeneity. However, moisture has no obvious effect on mixing homogeneity for a long mixing time. This nuanced understanding of the interplay between moisture, particle interactions, and mixing duration contributes valuable insights to optimize concrete mixing processes.

## 1. Introduction

The mixing of fresh concrete is a complicated process, involving the transformation of aggregate state. As moisture dynamically transports in the mixing process, the aggregate undergoes a transition from a dry to a wet state, leading to significant changes in the interaction between particles. Over the past few decades, numerous scholars have analyzed the complexities of particle interactions [1,2,3]. For dry particles, the primary interactions are transiently nonviscous inelastic collisions and friction. For wet particles, when the particles are close to each other, the liquid bridge is formed near the contact point, causing the particles to bond and altering the mechanical properties. It is noteworthy that even humidity in the air can contribute to cohesion by forming tiny liquid bridges at the points of contact between particles. Understanding the distinct nature of interactions for dry and wet particles is crucial in comprehending the intricate mechanics of concrete mixing. This knowledge lays the foundation for further advancements in optimizing concrete mixtures and the overall construction process.

The cohesion generated by the liquid bridge between particles has a great influence on stability [4,5,6], and even small amounts of liquid or vapor can contribute to form the liquid bridge [7,8,9]. The properties of wet granules are very different from dry granules. The presence of liquid not only affects the forces between particles but also leads to dynamic and static property changes. Tegzes et al. [10] investigated the effect of interstitial liquid on the physical properties of granular media. They measured the angle of repose as a function of liquid content and observed an increase in the angle of repose with higher liquid content. Samadani et al. [11] also studied the effect of fluid on the angle of repose of particles in silo, noting a dramatic increase in the angle of repose with a rising fluid content until it reached an ultimate value. Moisture also affects the mixing and segregation of wet particles. Samadani and Kudrolli [12] studied the effect of liquid on segregation by the image of the pile produced when particles were poured into the silo. They observed a substantial reduction in segregation with the addition of a small amount of liquid. Liao et al. [13] investigated the effect of liquids on particle segregation in drums and found that an increase in moisture content significantly affected the segregation of wet particles. Geromichalos et al. [14] investigated the effect of small amounts of liquid on the behavior of particles by shaking horizontally cylindrical jars containing glass beads with different liquid contents. They found that mixing homogeneity is highly dependent on liquid content, and a transition to viscoplastic behavior occurs at a critical liquid content. Zhu et al. [15] focused on the high-shear mixing process of a granular flow containing a high proportion of solid particles in a liquid. Their study demonstrated that a higher moisture content tends to slow down the mixing process. Overall, these studies collectively underscore the intricate influence of liquid content on the various physical properties and behaviors of granular materials.

In recent years, owing to the rapid development of computing technology, numerical simulation has become a prevalent method. The discrete element method (DEM), originally proposed by Cundall and Strack [16], stands out as a numerical method used to simulate the motion of discontinuous media. The liquid bridge force model proves instrumental in replicating the flow dynamics of wet particles [17]. Depending on the volume of the liquid, the interstitial liquid between particles exists in different states, which are pendular, funicular, and capillary [18]. Based on the DEM, Hassanzadeh et al. [19] studied the effect of cohesion on the macroscopic behavior of coarse particles through the angle of repose test. Additionally, a multitude of scholars employed DEM to explore the influence of moisture content on the mixing performance of aggregates [20,21]. Krenzer et al. [22] proposed a new DEM model to investigate the effect of moisture on the mixing process of fresh concrete. Examining the flow dynamics of dry and wet particles in a four-bladed mixer, Radl et al. [23] observed that the accumulation of wet particles was more pronounced compared to dry particles. Umer et al. [24] scrutinized the flow behavior of wet and dry particles on a single blade, noting the superior mixing performance of wet particles. Remy et al. [25] investigated the flow and agglomeration of wet particles in a bladed mixer using PIV and DEM. The results indicated that at a low moisture content, the convective and diffusive motion of wet particles intensified, enhancing mixing uniformity. However, at a higher moisture content, mixing uniformity diminished due to agglomeration. Shah [26] studied the flow of wet and dry particles in a vertical cylindrical mixer and revealed that the mixing efficiency of dry particles was higher than that of wet particles with higher moisture contents.

This paper investigated the effect of moisture content on the mixing homogeneity of sand and stone using the DEM. To capture the complex interactions between particles, the Hertz–Mindlin model was adopted for dry particles, while the linear cohesive model was utilized for wet particles. The moisture content was intricately linked to the parameters of the linear cohesive contact model. To determine the appropriate moisture content range for employing the constitutive models, experimental data on the angle of repose were meticulously fitted. This step helped establish guidelines for selecting the constitutive model based on the moisture content levels. Subsequently, the numerical method’s validity was assessed by comparing simulation results with experimental findings, ensuring the model’s accuracy in representing real granular behavior. Lastly, the mixing process of sand and pebbles across various moisture content variations was thoroughly analyzed. This investigation illustrates how moisture affects the homogeneity of the mixture, offering insights into optimizing mixing procedures for sand and stone compositions under different environmental conditions.

## 2. Contact Model

The interaction forces among particles play a crucial role in the behavior of granular materials, and these forces are significantly influenced by the moisture content of the particles. The distinction between dry and wet particles is evident in the nature of the forces at play. In the case of dry particles, there is no adhesive interaction between them. This lack of adhesive forces implies that the particles do not exhibit cohesion in their dry state. On the other hand, when particles are wet, cohesion arises due to the presence of liquid bridge forces. Liquid bridges can exist in various states, determined by the moisture content, and there are four primary types of liquid bridges: pendular, funicular, capillary, and slurry [27]. In the pendular state, particles are connected by a liquid bridge at the contact point. This type of bridge forms a pendular connection between adjacent particles. In the funicular state, some of the voids are completely filled with liquid, but there are still voids containing air. Liquid bridges generate around the contact points, enhancing adhesion between particles. In the capillary state, the liquid almost entirely fills the spaces between the particles, but the liquid on the surface of the particles is drawn back into the spaces under capillary action. This suction results in a cohesive interaction between particles. In the slurry state, particles are suspended in the fluid, the cohesion depends on the viscosity of the liquid rather than the liquid bridge force [28]. Understanding these different states of liquid bridges is crucial for comprehending the variations in cohesive forces and interactions between particles at different moisture levels. This knowledge is particularly valuable for the mixing homogeneity of sand and stone, where moisture content plays a key role in shaping the behavior of the particulate system.

### 2.1. Dry Particle Contact Model

For dry particles, the Hertz–Mindlin contact model is often used to represent their interaction. The normal force *F_n_* and tangential force *F_t_* are calculated as follows [29]:(1)$$F_{n}=\frac{4}{3}E^{*}\sqrt{R^{*}}{\delta_{n}}^{\frac{3}{2}}$$
(2)$$F_{t}=-S_{t}\delta_{t}$$
(3)$$\frac{1}{E^{*}}=\frac{\left(1-v_{i}^{2}\right)}{E_{i}}+\frac{\left(1-v_{j}^{2}\right)}{E_{j}}$$
(4)$$\frac{1}{R^{*}}=\frac{1}{R_{i}}+\frac{1}{R_{j}}$$
where *E^*^* is the equivalent Young’s modulus, *R^*^* is the equivalent radius, *ν_i_* and *ν_j_* are Poisson’s ratio, *E_i_* and *E_j_* are Young’s modulus, *R_i_* and *R_j_* are the radius of the particle, *S_t_* is the tangential stiffness, *δ_n_* is normal overlap and *δ_t_* is tangential overlap.

### 2.2. Wet Particle Contact Model

The liquid bridge force stands as one of the primary forces governing the interactions between wet particles. Notably, the cohesion engendered by a liquid bridge can surpass the gravitational force exerted by the particles’ own weight. This phenomenon underscores the significant role of liquid bridges in particle cohesion and aggregation. When the liquid films coating the surfaces of two particles come into contact, a liquid bridge is initiated. This bridge forms a bond between the particles, contributing to their cohesive behavior. Furthermore, following the collision and subsequent rebound of two particles, it is conceivable that a liquid bridge force persists within a certain distance between them [30,31]. Understanding these dynamics is pivotal for grasping the intricate interplay of forces within wet particulate systems. They elucidate the mechanisms behind particle cohesion and aggregation.

#### 2.2.1. Liquid Bridge Force

There are static and dynamic liquid bridge forces. The static liquid bridge force is determined by the combination of surface tension and hydrostatic pressure, while the dynamic liquid bridge force primarily results from the viscosity of the liquid filling the gap. Notably, when compared to the static counterpart, the dynamic liquid bridge force induced by the viscosity of water is usually considered negligible. In this paper, the focus is primarily on analyzing the static liquid bridge force. The process begins with the formation of a liquid film around the wet particle, uniformly covering their surfaces (refer to Figure 1). As the particles approach each other, a portion of the liquid film gradually merges, giving rise to the creation of a stable liquid bridge (illustrated in Figure 2). This phenomenon plays a crucial role in understanding particle cohesion and aggregation in wet conditions. In addition to the arc hypothesis, other assumptions are made in this paper: the surface of the liquid bridge is concave [32]; when the liquid bridge is stretched, the contact angle between the liquid bridge and the particle or wall is considered constant. The hysteresis phenomenon of the contact angle [33] is not considered, and the contact angle is assumed to be 0 degrees [34,35].

The premise of calculating the liquid bridge force is to determine the volume of liquid bridge according to the moisture content. The volume of liquid bridge *V* is calculated by Equation (5) [36].
(5)$$V=\frac{8\pi \rho_{p}{R^{3}}_{p}\omega_{0}}{3n_{c}\rho_{l}}$$
where *ρ_p_* is the particle density, *R_p_* is the particle radius, *ρ_l_* is the liquid density, *ω*_0_ is the moisture and *n_c_* is the coordination number, and in this paper *n_c_* is taken as 6 [37].

In this paper, the regression form of liquid bridge force *F* is adopted [17].
(6)$$F=2\pi R\gamma F^{*}$$
where *F^*^* is a dimensionless parameter and can be calculated by Equation (7).
(7)$$\ln F^{*}=f_{1}-f_{2}\exp(f_{3}\ln S^{+}+f_{4}\ln^{2}S^{+})$$
where *f*_1_, *f*_2_, *f*_3_, *f*_4_ are the coefficients [36].

The dimensionless parameter *S^+^* is defined by Equation (8).
(8)$$S^{+}=\frac{S}{\sqrt{V/R}}$$
where *S* is half of the distance of two particles; *R* is the average radius of curvature of the liquid bridge neck and is calculated by Equation (9).
(9)$$\frac{1}{R}=\frac{1}{2}(\frac{1}{R_{i}}+\frac{1}{R_{j}})$$

#### 2.2.2. Linear Cohesive Contact Model

The linear cohesive contact model employed in our study serves as a framework for characterizing the interaction between wet particles, specifically in scenarios where a liquid bridge is formed. This model is an extension of the Hertz–Mindlin contact model, incorporating a normal cohesive force to account for the effects of the liquid bridge. The mathematical representation of this model is expressed in Equation (10). This model provides a valuable tool for simulating and predicting the behavior of wet particles under various conditions, contributing to the broader understanding of particle–particle interactions in the presence of liquid bridges.
(10)F=kA
where *A* is the contact area of the particles and *k* is the cohesive energy density (J/m^3^).

## 3. Experiment

### 3.1. Angle of Repose Experiments

The equipment comprises a cylinder with a diameter of 104 mm and a height of 150 mm. Before the test, the cylinder was placed on a slab. After the aggregate was filled to the entire cylinder, it was lifted at a constant speed. Then, the aggregates collapsed under their own weight. The angle between the pyramidal surface and the horizontal plane is defined as the material angle of repose [38]. 

Two types of materials are used in the experiment: natural sand with a diameter up to 5 mm and pebbles ranging from 5 to 20 mm in diameter. The bulk density of the sand is 1618 kg/m^3^, while the pebbles have a bulk density of 1000 kg/m^3^. Both the sand and pebbles undergo a drying process in a dryer, with periodic weighing every hour until a constant mass is achieved. The mass at this point is considered the initial mass, representing the dry state with 0% moisture content. Following the initial weighing, the materials are soaked in water for 24 h. After removing excess water from the surface, the materials are weighed again. This mass is considered the saturated absorption water state mass, indicating the maximum water content. Subsequently, the soaked pebbles and sand are placed back into the dryer. At regular intervals (every 1 or 5 min), samples of pebbles or sand are taken out, weighed, and the moisture content was calculated. And then the angle of repose of the experiment is obtained. Due to the potential impact of external factors on the experiment, such as the lifting speed of the cylinder, the entire process is repeated three times for each moisture content. The results of the angle of repose at both the dry and saturated water states are presented in Figure 3. The state is obviously different for the sand.

#### 3.1.1. Relationship between Angle of Repose and Moisture of Pebbles

The experimental results of the angle of repose of the pebbles at different moisture contents are shown in Table 1. The variation in the angle of repose with moisture content is shown in Figure 4. The experimental data were fitted, and the fitting equation is shown in Equation (11), and the fitting curve is shown in Figure 4. The correlation coefficient R^2^ is 0.9735.
(11)$$\theta_{p}=\frac{q_{1}\omega_{p}^{5}+q_{2}\omega_{p}^{4}+q_{3}\omega_{p}^{3}+q_{4}\omega_{p}^{2}+q_{5}\omega_{p}+q_{6}}{p_{1}\omega_{p}^{4}+p_{2}\omega_{p}^{3}+p_{3}\omega_{p}^{2}+p_{4}\omega_{p}+p_{5}}$$
where *ω_p_* is the moisture content of sand, and the coefficients *q*_1_~*q*_5_, *p*_1_~*p*_5_ are
$$q_{1}=-9.4400\times 10^{8}\;\;q_{2}=6.5181\times 10^{7}\;\;q_{3}=2.8968\times 10^{5}\;\;q_{4}=-6178.4091\;q_{5}=-10.8606\;\;q_{6}=0.1594\;\;p_{1}=2.3825\times 10^{6}\;\;p_{2}=3795.581\;p_{3}=-84.3253\;\;p_{4}=-1.7183\;\;p_{5}=0.01$$


#### 3.1.2. Relationship between Angle of Repose and Moisture of Sand

The experimental results of the angle of repose of the sand are shown in Table 2. The experimental data were fitted, and the fitting result is shown in Equation (12), and the curve is shown in Figure 5. The correlation coefficient R^2^ is 0.9756.
(12)$$\theta_{s}=\frac{q_{1}{\omega_{s}}^{5}+q_{2}{\omega_{s}}^{4}+q_{3}{\omega_{s}}^{3}+q_{4}{\omega_{s}}^{2}+q_{5}\omega_{s}+q_{6}}{{\omega_{s}}^{4}+p_{1}{\omega_{s}}^{3}+p_{2}{\omega_{s}}^{2}+p_{3}\omega_{s}+p_{4}}$$
where *ω_s_* is the moisture content of sand, and the coefficients in Equation (12) are
$$q_{1}=-20639.146\;\;q_{2}=-1857.9775\;\;q_{3}=827.1269\;\;q_{4}=-95.8706\;q_{5}=6.6125\;\;q_{6}=0.2319\;\;p_{1}=-11.6050\;\;p_{2}=-0.0807\;p_{3}=0.1398\;\;p_{4}=0.01$$


Figure 4 and Figure 5 illustrate the correlation between the angle of repose and moisture content. Initially, the angle of repose rises with increasing moisture content, but then it declines once the moisture content surpasses a critical threshold. This phenomenon occurs because, as moisture content increases, the state of the liquid bridges between particles transitions from pendular to slurry. Consequently, cohesion, generated by these liquid bridges, gradually escalates from zero to its peak, before eventually dwindling to zero in the slurry state. In view of this, the wet particle model is applicable when the moisture content ranges from 0% to 0.75% for pebbles and from 0% to 10.9% for sand. Notably, the steepness of the curve varies; it is particularly pronounced when the moisture content of pebbles hovers around 0% to 0.4%, and that of sand is roughly 0% to 1.1%. During this interval, the angle of repose shows a positive correlation with moisture content, indicative of the pendular state between particles.

As the moisture content of pebbles reaches 0.4% to 0.55% and sand reaches approximately 1.1% to 5.2%, the angle of repose stabilizes. Within this range, the number of contact points between particles maximizes, indicating a transition to the funicular state. Here, the suction force influenced by the liquid content is lower than that in the pendular state [28]. The curve then exhibits a sharp ascent once more at moisture levels of 0.55% to 0.75% for pebbles and around 5.2% to 10.9% for sand. During this phase, the angle of repose increases at an accelerated rate with rising moisture content, with even minor increments leading to noticeable changes. At this stage, the liquid bridges between particles shift from funicular to capillary, filling all interstices between particles and enhancing cohesion. However, beyond these moisture thresholds, the curves demonstrate a decreasing trend when the moisture contents of pebbles and sand are greater than 0.75% and 10.9%, respectively. At this phase, the type of liquid bridge between particles becomes slurry. The number of liquid bridges decreases and the cohesion decreases, so the angle of repose decreases.

### 3.2. Sand and Pebble Mixing Experiment

#### 3.2.1. Material 

In total, 44.6 kg dry pebbles and 25.6 kg dry sand were utilized in the experiment. Consistent with Section 3.1, pebbles and sand with different moisture contents were obtained, as shown in Table 3. A mixing machine was used to mix the materials. The pebbles and sand were put into the mixer in turn to mix for 30 s.

#### 3.2.2. Sampling Method

In the mixing process, assessing the uniformity of the mixture is crucial, and this is often performed through calculations using data obtained from sampling. Due to the challenges of sampling directly from the mixing machine, a sampler (Figure 6) was employed. This sampler was equipped with 20 meshes with 612 mm × 610 mm in length and weight. After the mixing process, the aggregates were discharged into this sampler. The standard deviation of mixing homogeneity (*S_mh_*), as expressed in Equation (13), serves as the chosen mixing index for evaluating the quality of the mixing. The use of standard deviation in this context is indicative of the degree of variation in or dispersion of the properties being measured, providing insights into the homogeneity of the mixed materials.
(13)$$S_{mh}=\sqrt{\frac{1}{n-1}\sum_{i=1}^{n}{\left(X_{i}-\bar{X}\right)}^{2}}$$
(14)$$\bar{X}=\frac{1}{n}\sum_{i=1}^{n}X_{i}$$where *n* is total number of samples, and $\bar{X}$ is the average value of the mass ratio of pebbles to the sample.

#### 3.2.3. Result of Mixing Experiment

The experimental results presented in Figure 7 reveal a consistent pattern: regardless of the moisture content, the aggregates at the middle position exhibit higher quantities compared to other locations. To further investigate and quantify mixing homogeneity, a quantitative analysis was conducted by calculating the standard deviation, as outlined in Table 4. The findings in Table 4 demonstrate that the standard deviation of the mixing degree tends to be higher when the material has elevated moisture content or is completely dry, in contrast to situations with less moisture content. This observation implies that the homogeneity of the mixture is enhanced when particles possess less moisture content [25]. In other words, the variability in the distribution of aggregates is more pronounced when the material is either excessively wet or completely dry, suggesting that maintaining a moderate moisture level contributes to better mixing homogeneity. 

## 4. Simulation

The particle shape was simplified in the EDEM simulation, and spherical particles with diameters of 5 mm and 12 mm were employed to represent sand and pebbles, respectively. According to the moisture content, the volume of liquid bridge is calculated by Equation (5). Then, the liquid bridge force *F* is calculated based on the liquid bridge volume (Equation (6)). Subsequently, the cohesive density parameter *k* can be calculated by Equation (10). The diameter of the cylinder used in the simulation is 104 mm, and the cylinder moved vertically upwards at a velocity of 200 mm/s until all the particles stopped moving, at which point the simulation finished.

### 4.1. Parameter Setting 

The properties of sand, pebble particles, and wall are shown in Table 5, and the contact parameters are shown in Table 6.

### 4.2. Simulation of Angle of Repose 

#### 4.2.1. Simulation of Pebble Angle of Repose 

After determining the simulation parameters, angle of repose simulations were carried out using EDEM, and the final stacking shapes are shown in Figure 8. And the results are shown in Table 7 and Figure 9. The relative error is within 5%.

#### 4.2.2. Simulation of Sand Angle of Repose 

Figure 10 displays the final stacking pattern of the sand. The corresponding results are summarized in Table 8, while Figure 11 offers additional insights into the comparative analysis between the simulation and experimental outcomes. Upon examination of the data presented in Table 8 and Figure 11, it becomes evident that the maximum relative error between the simulation and experimental results does not surpass 5%. This indicates a high level of agreement between the simulated stacking pattern and the actual experimental observations, validating the accuracy and reliability of the simulation methodology utilized in this study.

### 4.3. Simulation of Mixing of Sand and Pebbles 

In the conducted simulation of three series of mixing processes, the moisture content of both pebbles and sand is detailed in Table 3. The cohesive energy density parameter (*k*) for the linear cohesive contact model is computed using Equation (10) and is detailed in Table 9. The mixing process is visually depicted in Figure 12, where 2144 pebble particles are initially generated at the bottom of the mixer, followed by the creation of 19,862 sand particles on top of the pebbles. At 7 s, the mixer initiates rotation with a rotational velocity of 48 rpm for 25 s. At 28 s, the door of the mixer is opened, and the sand and pebbles are discharged into the sampler. The discharging process is illustrated in Figure 13. The sampling results are shown in Figure 14.

The comparison of the simulation and experiment of the pebble mass ratio is shown in Figure 15. It can be noticed that larger errors occur in the outermost grids, such as those numbered 17 and 18. This is due to the fact that there are fewer particles and less mass in the peripheral grid compared to the other grids. Table 10 shows that the relative errors between the experiments and simulations are all within 5%.

### 4.4. Effect of Mixing Time on Homogeneity

The standard deviation of the mixing homogeneity curve for sand and pebbles with different moisture contents in the mixer is shown in Figure 16. The mixing process exhibits three distinct phases: a fast mixing phase, a slow mixing phase, and a dynamic equilibrium phase. Blade mixing initiates at 7 s, and the standard deviation of the mixing degree decreases rapidly. The slow mixing phase spans from 13 to 18 s, followed by a dynamic equilibrium phase where the standard deviation of mixing homogeneity stabilizes around 0.05. Notably, these findings align with the trends observed in the standard deviation of the mixing degree reported by Schmelzle [39] and Zuo [40]. 

During 8–14 s, the standard deviation of the mixing degree for dry particles significantly surpasses that of wet particles. This suggests that a small amount of liquid enhances homogeneity within a short mixing time. However, as mixing time extends, the distinction in standard deviation between the mixing degrees of dry and wet particles diminishes. This observation holds significance for concrete production in mixing plants where limited mixing time is available due to energy consumption constraints.

## 5. Conclusions

In this paper, the effect of moisture content on the sand and pebble mixing process was investigated using the DEM. Depending on the moisture content, either a dry particle contact model or a wet particle contact model was selected. Dry particle interactions were characterized by the Hertz–Mindlin model, while wet particle interactions were described using the linear cohesive contact model. The following conclusions were obtained.

The parameters of the linear cohesive contact model were defined as a function of the liquid bridge force. This study observed that the angle of repose exhibited an initial increase, followed by a decrease, with varying moisture content. Specifically, when the moisture content of the stone was within the range of 0% to 0.75% and that of the sand was within the range of 0% to 10.9%, the wet contact model was employed. Furthermore, this paper highlighted that the errors between the experimental and simulation results were within 5%, demonstrating the reliability of the numerical method used.

The mixing process of sand and pebbles was characterized by three phases: the fast mixing phase, the slow mixing phase, and the dynamic equilibrium phase. During short mixing times, the standard deviation of the mixing homogeneity of wet particles was found to be lower than that of dry particles. This observation suggests that the presence of a small amount of liquid enhances mixing homogeneity in the early stages of the mixing process.

## Figures and Tables

**Figure 1 materials-17-00898-f001:**
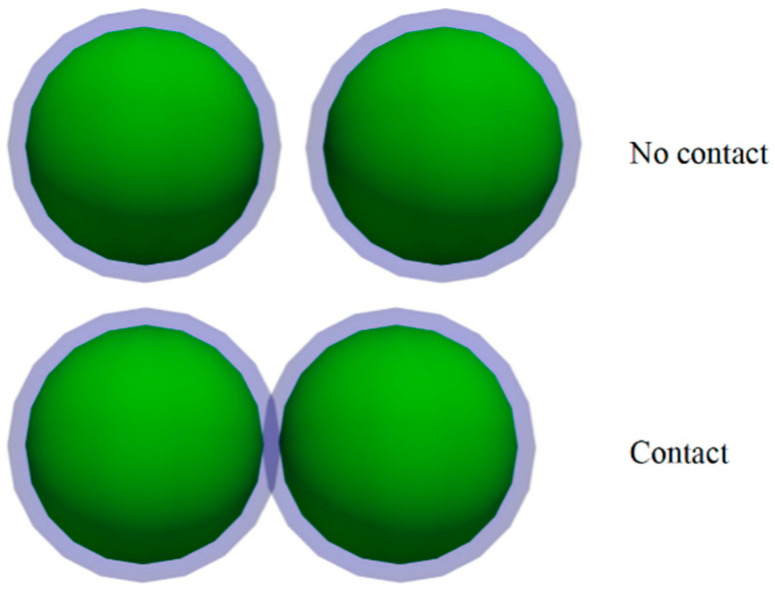
Type of wet particle contact.

**Figure 2 materials-17-00898-f002:**
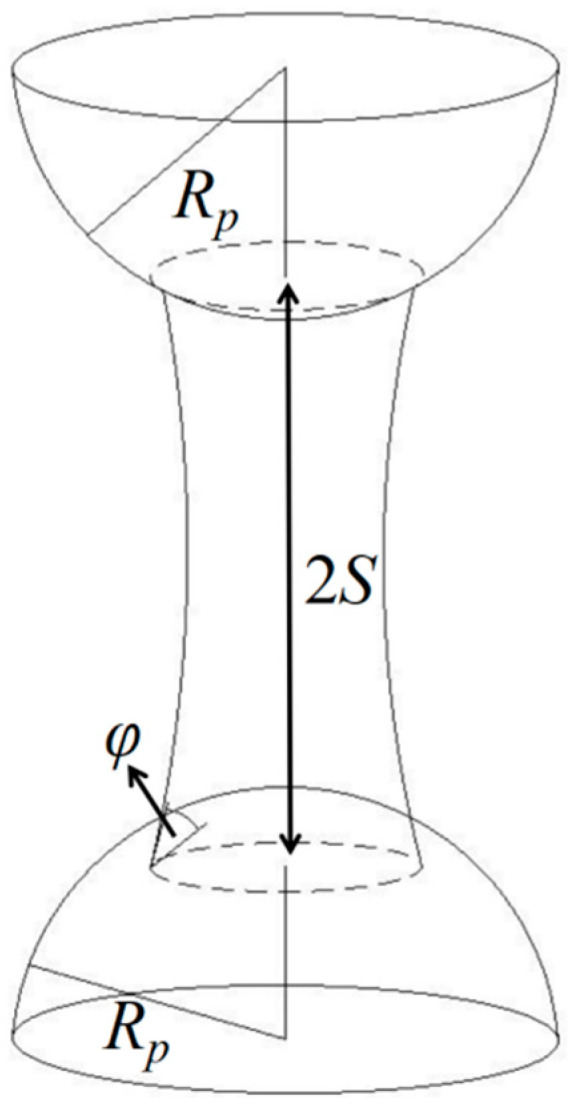
Liquid bridge model.

**Figure 3 materials-17-00898-f003:**
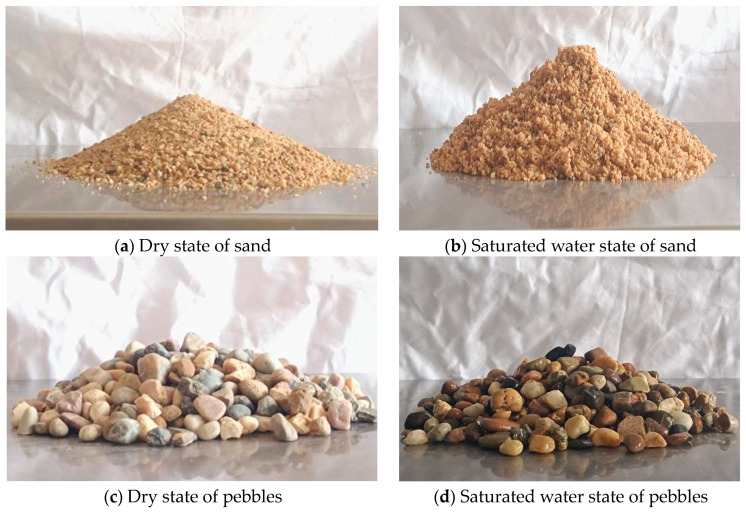
Final state of angle of repose.

**Figure 4 materials-17-00898-f004:**
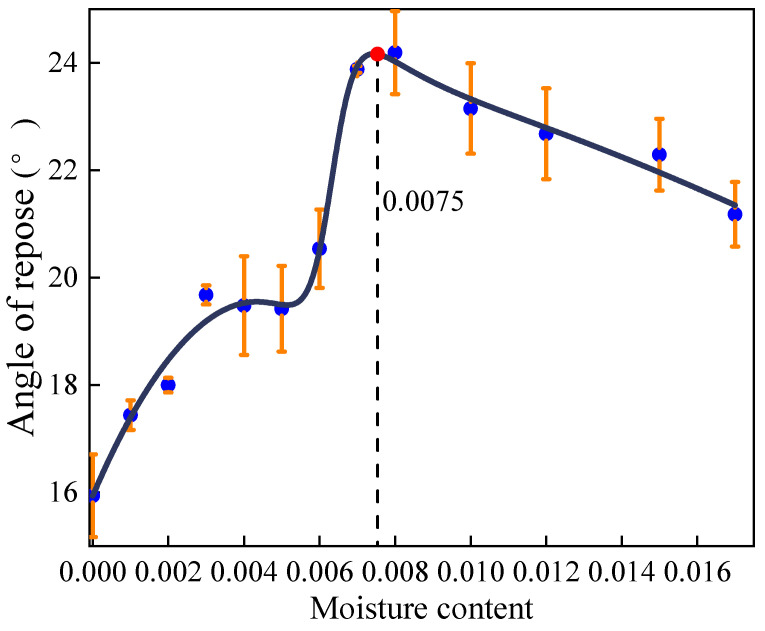
Fitting curve of pebbles’ moisture content and angle of repose.

**Figure 5 materials-17-00898-f005:**
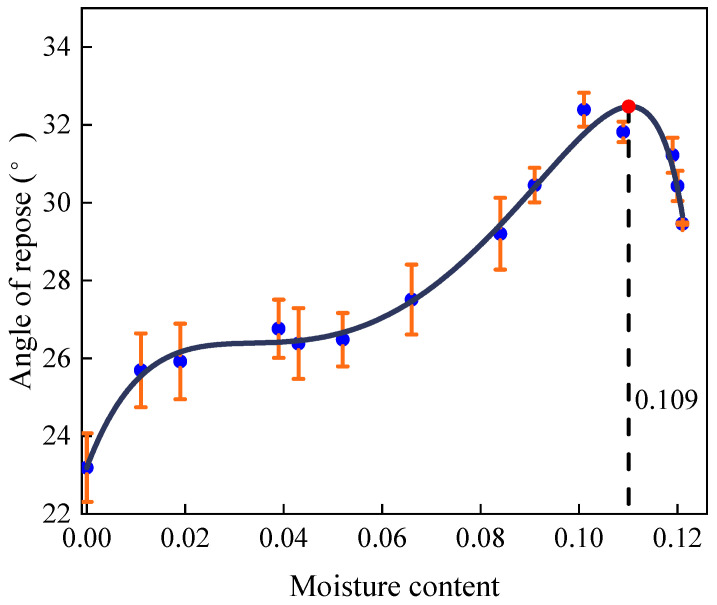
Fitting curve of sand’s moisture content and angle of repose.

**Figure 6 materials-17-00898-f006:**
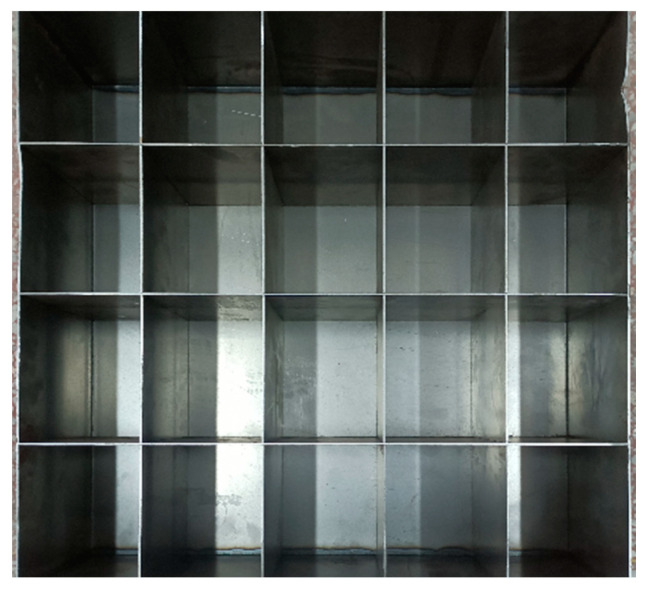
Sampler.

**Figure 7 materials-17-00898-f007:**
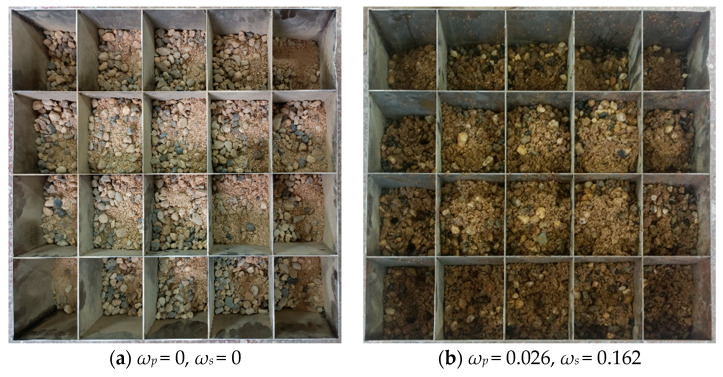
Sampling results for different moisture contents.

**Figure 8 materials-17-00898-f008:**
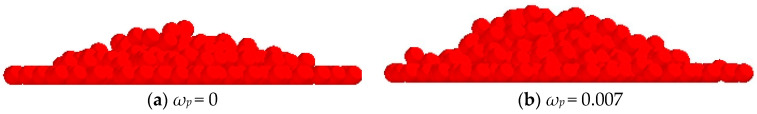
Final stacking pattern of pebbles.

**Figure 9 materials-17-00898-f009:**
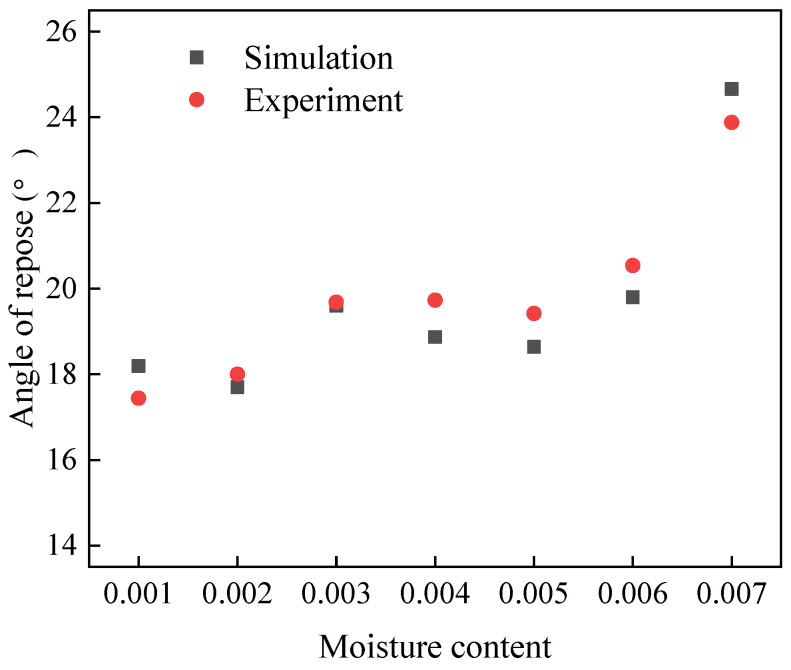
Comparison of experimental and simulating pebble angle of repose.

**Figure 10 materials-17-00898-f010:**
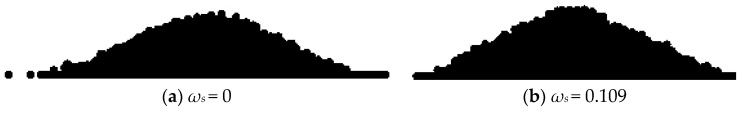
Final stacking pattern of the sand.

**Figure 11 materials-17-00898-f011:**
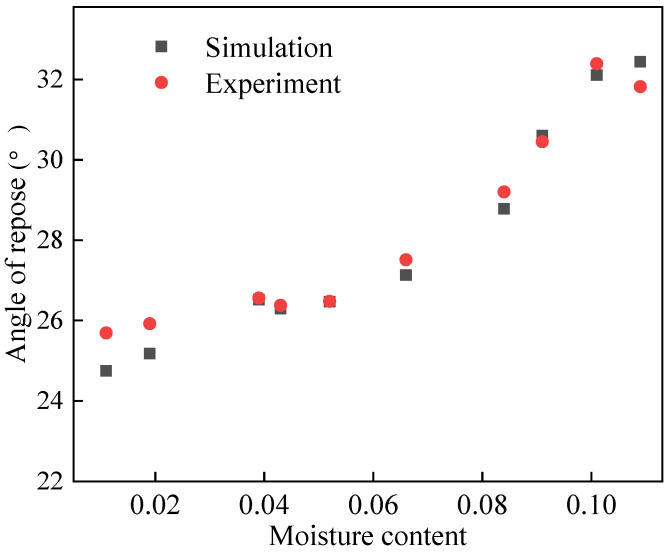
Comparison of sand angle of repose simulation and experiment.

**Figure 12 materials-17-00898-f012:**
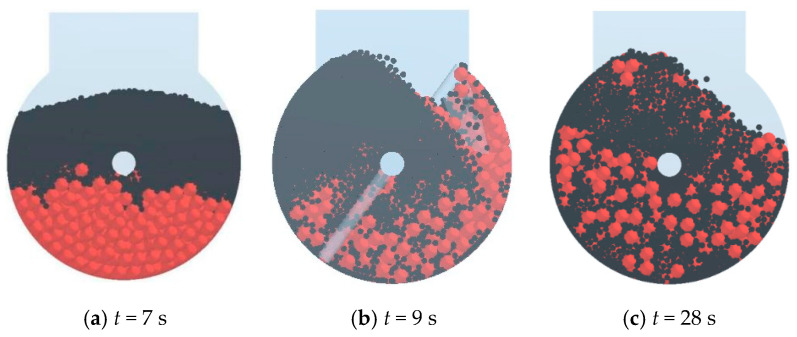
Mixing process of sand and pebbles.

**Figure 13 materials-17-00898-f013:**
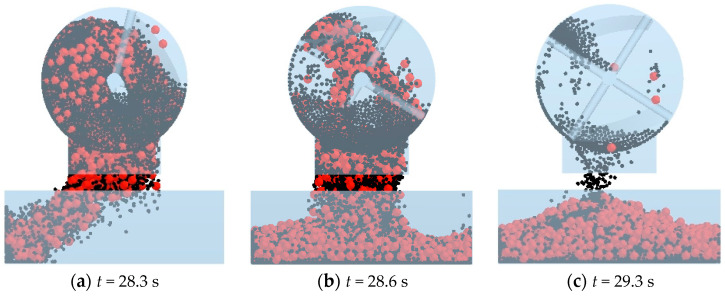
Discharging process.

**Figure 14 materials-17-00898-f014:**
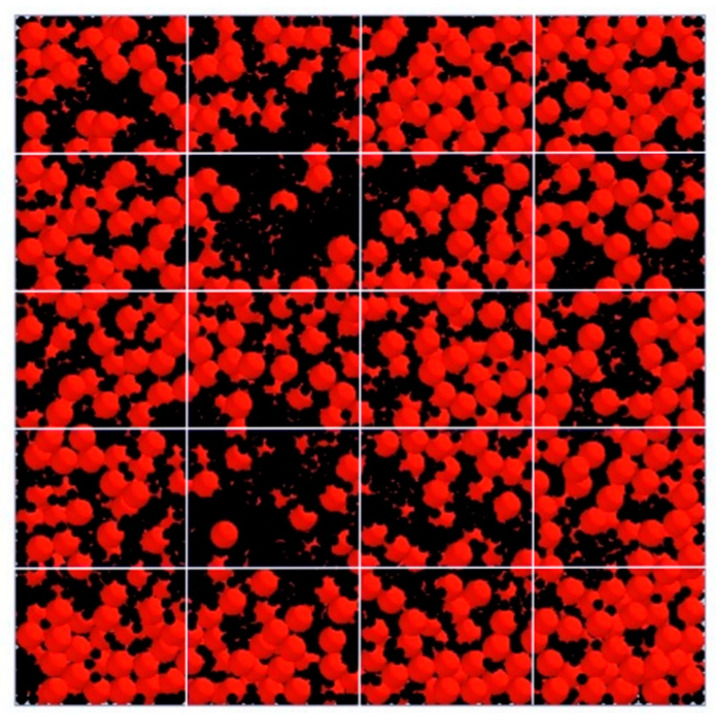
Sampling results after discharge.

**Figure 15 materials-17-00898-f015:**
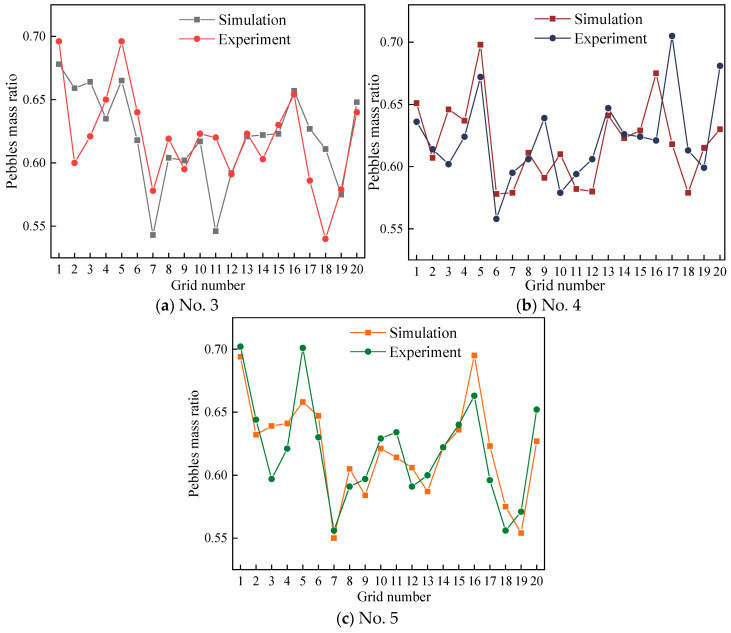
Comparison of simulation and experiment.

**Figure 16 materials-17-00898-f016:**
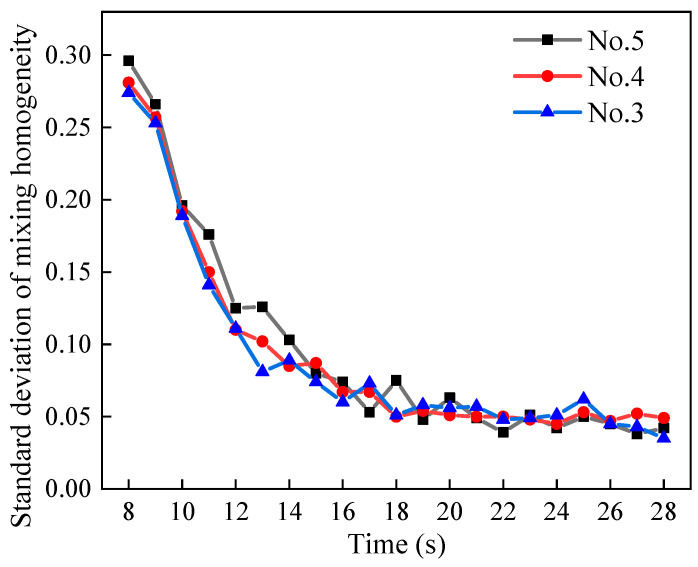
Standard deviation of mixing homogeneity for different moisture contents.

**Table 1 materials-17-00898-t001:** Experimental results of pebble angle of repose.

Moisture Content	Experiment Value (°)			Average Value (°)
	1	2	3	
0	15.07	16.22	16.53	15.94
0.001	17.24	19.95	17.63	17.44
0.002	17.9	20	18.09	18.00
0.003	19.8	22.66	19.55	19.68
0.004	18.56	20.4	19.49	19.48
0.005	18.69	20.27	19.29	19.42
0.006	20.02	21.39	20.81	20.54
0.007	20.38	23.92	23.83	23.88
0.008	23.3	24.59	24.68	24.19
0.010	22.21	23.39	23.84	23.15
0.012	21.72	23.03	23.3	22.68
0.015	21.53	22.58	22.77	22.29
0.017	20.75	24.03	21.6	21.18

**Table 2 materials-17-00898-t002:** Experimental results of the sand’s angle of repose.

Moisture Content	Experiment Value (°)			Average Value (°)
	1	2	3	
0	23.61	22.18	23.79	23.19
0.011	25.36	24.95	26.76	25.69
0.019	25.8	25.02	26.95	25.92
0.039	26.45	26.21	27.61	26.76
0.043	25.86	25.64	27.65	26.38
0.052	26.21	25.97	27.26	26.48
0.066	26.92	27.07	28.55	27.51
0.084	28.18	29.98	29.44	29.20
0.091	30.96	30.19	30.19	30.45
0.101	32.44	32.8	31.93	32.39
0.109	32.01	31.93	31.52	31.82
0.119	31.5	31.47	30.7	31.22
0.120	30.66	30.64	29.98	30.43
0.121	29.44	29.44	29.49	29.46

**Table 3 materials-17-00898-t003:** Moisture content of sand and pebbles.

Series No.	Drying Time (min.)	Pebbles’ Moisture Content	Sand Moisture Content
1	0	0.026	0.162
2	10	0.009	0.127
3	20	0.005	0.084
4	30	0.001	0.044
5	-	0	0

**Table 4 materials-17-00898-t004:** Mixing experiment results.

	Drying Time				
	0 min. (Saturated State)	10 min.	20 min.	30 min.	24 h. (Dry State)
*S_mh_*	0.042	0.037	0.038	0.035	0.041

**Table 5 materials-17-00898-t005:** Properties of raw material.

Material	Poisson’s Ratio	Density (kg/m^3^)	Shear Modulus (Pa)
Pebble	0.35	2680	8 × 10^9^
Sand	0.25	2500	1 × 10^8^
Wall	0.30	7850	1 × 10^10^

**Table 6 materials-17-00898-t006:** Contact parameters.

Type of Contact	Coefficient of Restitution	Coefficient of Static Friction	Coefficient of Rolling Friction
Pebble–Pebble	0.10	0.20	0.10
Sand–Sand	0.05	0.32	0.15
Pebble–Wall	0.10	0.20	0.18
Sand–Wall	0.05	0.17	0.10

**Table 7 materials-17-00898-t007:** Simulation result of pebble angle of repose.

Moisture Content	Cohesive Density (10^7^ J/m^3^)	Simulation Value (°)	Experiment Value (°)	Relative Error (%)
0	0	15.51	16.07	−3.48
0.001	1.94	18.19	17.44	4.30
0.002	2.06	17.70	18.00	−1.67
0.003	2.12	19.60	19.68	−0.41
0.004	2.15	18.87	19.73	−4.36
0.005	2.16	18.64	19.42	−4.02
0.006	2.22	19.80	20.54	−3.60
0.007	2.43	24.66	23.88	3.27

**Table 8 materials-17-00898-t008:** Simulation result of sand angle of repose.

Moisture Content	Cohesive Density (10^5^ J/m^3^)	Simulation Value (°)	Experiment Value (°)	Relative Error (%)
0	0.00	23.06	23.19	−0.56
0.011	0.29	24.75	25.69	−3.66
0.019	0.38	25.18	25.92	−2.85
0.039	0.42	26.52	26.56	−0.15
0.043	0.43	26.30	26.38	−0.30
0.052	0.46	26.47	26.48	−0.04
0.066	0.60	27.13	27.51	−1.38
0.084	0.91	28.78	29.2	−1.44
0.091	1.06	30.60	30.45	0.49
0.101	1.27	32.11	32.39	−0.86
0.109	1.38	32.44	31.82	1.95

**Table 9 materials-17-00898-t009:** Cohesive energy density *k*.

Series No.	*k*_p_ (10^7^ J/m^3^)	*k*_s_ (10^4^ J/m^3^)	*k*_p_ − *k*_s_ (10^6^ J/m^3^)
3	2.16	9.12	2.60
4	1.95	4.32	2.30
5	0	0	0

**Table 10 materials-17-00898-t010:** Simulation and experiment standard deviation of mixing.

Series No.	Simulating *S_mh_*	Experimental *S_mh_*	Relative Error (%)
3	0.0372	0.0382	−2.61
4	0.0333	0.0349	−4.58
5	0.0388	0.0408	−4.90

## Data Availability

Data are contained within the article.

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
