# Peer review of "Numerical Study on Effect of Aggregate Moisture on Mixing Process"

_materials, 2024, doi:10.3390/ma17040898_

Round 1

Reviewer 1 Report

Comments and Suggestions for Authors

Comments as attached.

This manuscript aims to investigate the effect of aggregate's moisture on the mixing process of sand and stone. The interaction between dry particles was described by the Hertz-Mindlin model, while the interaction between wet particles was calculated by the linear cohesion model with considering the liquid bridge force. The functional relationship between moisture and linear cohesive contact model parameter was established. The results show that the numerical method can be used to simulate the mixing process. When the moisture content of the pebble is between 0% and 0.75% and that of sand is between 0% and 10.9%, the linear cohesion model is adopted. The standard deviation of the mixing degree of wet particles is lower than that of dry particles for short mixing time, indicating that a small amount of liquid enhances the mixing homogeneity. Overall, the manuscript provide a better understanding the effect of aggregate moisture on mixing processs using numerical study. 

Comments on the Quality of English Language

appropriate for publication with minor correction.

Reviewer 2 Report

Comments and Suggestions for Authors

In the section 3.1, row 149, how did the authors verify the reaching of a s.s.d. (saturated surface dry) condition? Please explain. The European standard EN 1097-6:2013 provides an exhaustive guidance for the determination of saturated surface dry condition for both sand and gravels. 2. In Figure 4 and 5, please describe where the s.s.d. moisture condition is located. The determination of such a point in these curves would be of great research interest. 3. From section 4.3, why did the authors use the nomenclature series no. 3,4 and 5? Are 1 and 2 missed? Please explain, otherwise the names for such series should be adjusted.

Comments on the Quality of English Language

None

Reviewer 3 Report

Comments and Suggestions for Authors

This paper investigates the impact of aggregate moisture on the mixing process of sand and stone using the DEM method. The interaction between dry particles was described using the Herd-Mindlin model, while the interaction between wet particles was calculated using the linear cohesion model. The study found that the linear cohesion model can be used to simulate the mixing process when the moisture content of pebble is between 0% and 0.75% and sand is between 0% and 10.9%.  The author concluded that moisture doesn't significantly affect mixing homogeneity for long mixing times. The paper fits within the topics of the journal and the subject may be of interest of its readers. However, the current level of the manuscript does not allow its recommendation for publication. The special comments are given below

-        The used references are old and new research works must be analyzed and cited

-        A proof reading is necessary because some typographic and grammatical errors exit

-        Explain the abbreviation DEM in the abstract

-        The introduction is very vague and a more detailed analysis of the literature is necessary. Authors must position the work more clearly by highlighting the added value

-        Error bars must be added on all the figures representing average data

-        How the authors justify the fact that the hysteresis phenomenon of the contact angle and the contact angel can be neglected without any effect on the results

-        Why were simple compactness tests not done to reinforce the results?

Comments on the Quality of English Language

A proof reading is necessary because some typographic and grammatical errors exit

Round 2

Reviewer 3 Report

Comments and Suggestions for Authors

The authors successfully adressed all the reviewer's remark. The revised manuscript can be accepted